# RECASTING GRADIENT-BASED META-LEARNING AS HIERARCHICAL BAYES

**Erin Grant**[12], **Chelsea Finn**[12], **Sergey Levine**[12], **Trevor Darrell**[12], **Thomas Griffiths**[13]

[1] Berkeley AI Research (BAIR), University of California, Berkeley
[2] Department of Electrical Engineering & Computer Sciences, University of California, Berkeley
[3] Department of Psychology, University of California, Berkeley

`{eringrant,cbfinn,svlevine,trevor,tom_griffiths}@berkeley.edu`

## ABSTRACT

*Meta-learning* allows an intelligent agent to leverage prior learning episodes as a basis for quickly improving performance on a novel task. Bayesian hierarchical modeling provides a theoretical framework for formalizing meta-learning as inference for a set of parameters that are shared across tasks. Here, we reformulate the model-agnostic meta-learning algorithm (MAML) of Finn et al. (2017) as a method for probabilistic inference in a hierarchical Bayesian model. In contrast to prior methods for meta-learning via hierarchical Bayes, MAML is naturally applicable to complex function approximators through its use of a scalable gradient descent procedure for posterior inference. Furthermore, the identification of MAML as hierarchical Bayes provides a way to understand the algorithm's operation as a meta-learning procedure, as well as an opportunity to make use of computational strategies for efficient inference. We use this opportunity to propose an improvement to the MAML algorithm that makes use of techniques from approximate inference and curvature estimation.

## 1 INTRODUCTION

A remarkable aspect of human intelligence is the ability to quickly solve a novel problem and to be able to do so even in the face of limited experience in a novel domain. Such fast adaptation is made possible by leveraging prior learning experience in order to improve the efficiency of later learning. This capacity for *meta-learning* also has the potential to enable an artificially intelligent agent to learn more efficiently in situations with little available data or limited computational resources (Schmidhuber, 1987; Bengio et al., 1991; Naik & Mammone, 1992).

In machine learning, meta-learning is formulated as the extraction of domain-general information that can act as an inductive bias to improve learning efficiency in novel tasks (Caruana, 1998; Thrun & Pratt, 1998). This inductive bias has been implemented in various ways: as learned hyperparameters in a hierarchical Bayesian model that regularize task-specific parameters (Heskes, 1998), as a learned metric space in which to group neighbors (Bottou & Vapnik, 1992), as a trained recurrent neural network that allows encoding and retrieval of episodic information (Santoro et al., 2016), or as an optimization algorithm with learned parameters (Schmidhuber, 1987; Bengio et al., 1992).

The model-agnostic meta-learning (MAML) of Finn et al. (2017) is an instance of a learned optimization procedure that directly optimizes the standard gradient descent rule. The algorithm estimates an initial parameter set to be shared among the task-specific models; the intuition is that gradient descent from the learned initialization provides a favorable inductive bias for fast adaptation. However, this inductive bias has been evaluated only empirically in prior work (Finn et al., 2017).

In this work, we present a novel derivation of and a novel extension to MAML, illustrating that this algorithm can be understood as inference for the parameters of a prior distribution in a hierarchical Bayesian model. The learned prior allows for quick adaptation to unseen tasks on the basis of an implicit predictive density over task-specific parameters. The reinterpretation as hierarchical Bayes gives a principled statistical motivation for MAML as a meta-learning algorithm, and sheds light on the reasons for its favorable performance even among methods with significantly more parameters.

More importantly, by casting gradient-based meta-learning within a Bayesian framework, we are able to improve MAML by taking insights from Bayesian posterior estimation as novel augmentations to the gradient-based meta-learning procedure. We experimentally demonstrate that this enables better performance on a few-shot learning benchmark.

## 2    META-LEARNING FORMULATION

The goal of a meta-learner is to extract task-general knowledge through the experience of solving a number of related tasks. By using this learned prior knowledge, the learner has the potential to quickly adapt to novel tasks even in the face of limited data or limited computation time.

Formally, we consider a dataset $\mathscr{D}$ that defines a distribution over a family of tasks $\mathcal{T}$. These tasks share some common structure such that learning to solve a single task has the potential to aid in solving another. Each task $\mathcal{T}$ defines a distribution over data points $\mathbf{x}$, which we assume in this work to consist of inputs and either regression targets or classification labels $\mathbf{y}$ in a supervised learning problem (although this assumption can be relaxed to include reinforcement learning problems; *e.g.,* see Finn et al., 2017). The objective of the meta-learner is to be able to minimize a task-specific performance metric associated with any given unseen task from the dataset given even only a small amount of data from the task; *i.e.,* to be capable of fast adaptation to a novel task.

In the following subsections, we discuss two ways of formulating a solution to the meta-learning problem: gradient-based hyperparameter optimization and probabilistic inference in a hierarchical Bayesian model. These approaches were developed orthogonally, but, in Section 3.1, we draw a novel connection between the two.

### 2.1    META-LEARNING AS GRADIENT-BASED HYPERPARAMETER OPTIMIZATION

A parametric meta-learner aims to find some shared parameters $\boldsymbol{\theta}$ that make it easier to find the right task-specific parameters $\boldsymbol{\phi}$ when faced with a novel task. A variety of meta-learners that employ gradient methods for task-specific fast adaptation have been proposed (*e.g.,* Andrychowicz et al., 2016; Li & Malik, 2017a;b; Wichrowska et al., 2017). MAML (Finn et al., 2017) is distinct in that it provides a gradient-based meta-learning procedure that employs a single additional parameter (the meta-learning rate) and operates on the same parameter space for both meta-learning and fast adaptation. These are necessary features for the equivalence we show in Section 3.1.

To address the meta-learning problem, MAML estimates the parameters $\boldsymbol{\theta}$ of a set of models so that when one or a few batch gradient descent steps are taken from the initialization at $\boldsymbol{\theta}$ given a small sample of task data $\mathbf{x}_{j_1}, \ldots, \mathbf{x}_{j_N} \sim p_{\mathcal{T}_j}(\mathbf{x})$ each model has good generalization performance on another sample $\mathbf{x}_{j_{N+1}}, \ldots, \mathbf{x}_{j_{N+M}} \sim p_{\mathcal{T}_j}(\mathbf{x})$ from the same task. The MAML objective in a maximum likelihood setting is

$$\mathcal{L}(\boldsymbol{\theta}) = \frac{1}{J} \sum_j \left[ \frac{1}{M} \sum_m - \log p\big( \mathbf{x}_{j_{N+m}} \mid \underbrace{\boldsymbol{\theta} - \alpha \, \nabla_{\boldsymbol{\theta}} \frac{1}{N} \sum_n - \log p\left( \mathbf{x}_{j_n} \mid \boldsymbol{\theta} \right)}_{\phi_j} \big) \right] \qquad (1)$$

where we use $\phi_j$ to denote the updated parameters after taking a single batch gradient descent step from the initialization at $\boldsymbol{\theta}$ with step size $\alpha$ on the negative log-likelihood associated with the task $\mathcal{T}_j$. Note that since $\phi_j$ is an iterate of a gradient descent procedure that starts from $\boldsymbol{\theta}$, each $\phi_j$ is of the same dimensionality as $\boldsymbol{\theta}$. We refer to the inner gradient descent procedure that computes $\phi_j$ as *fast adaptation*. The computational graph of MAML is given in Figure 1 (left).

### 2.2    META-LEARNING AS HIERARCHICAL BAYESIAN INFERENCE

An alternative way to formulate meta-learning is as a problem of probabilistic inference in the hierarchical model depicted in Figure 1 (right). In particular, in the case of meta-learning, each task-specific parameter $\phi_j$ is distinct from but should influence the estimation of the parameters $\{\phi_{j'} \mid j' \neq j\}$ from other tasks. We can capture this intuition by introducing a meta-level parameter $\boldsymbol{\theta}$ on which each task-specific parameter is statistically dependent. With this formulation, the mutual dependence of the task-specific parameters $\phi_j$ is realized only through their individual dependence

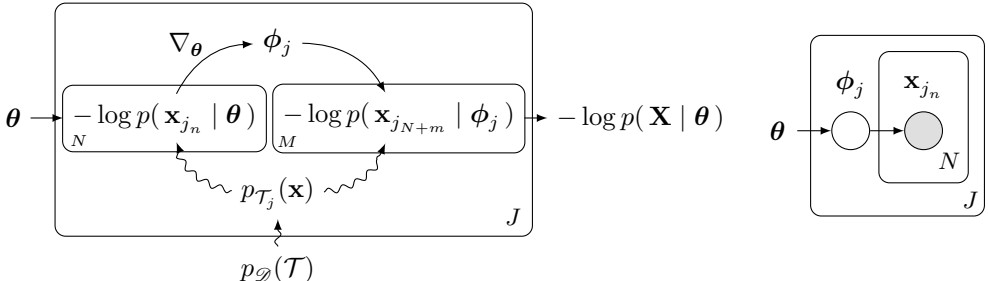

**Figure 1:** **(Left)** The computational graph of the MAML (Finn et al., 2017) algorithm covered in Section 2.1. Straight arrows denote deterministic computations and crooked arrows denote sampling operations. **(Right)** The probabilistic graphical model for which MAML provides an inference procedure as described in Section 3.1. In each figure, plates denote repeated computations (**left**) or factorization (**right**) across independent and identically distributed samples.

on the meta-level parameters $\boldsymbol{\theta}$. As such, estimating $\boldsymbol{\theta}$ provides a way to constrain the estimation of each of the $\phi_j$.

Given some data in a multi-task setting, we may estimate $\boldsymbol{\theta}$ by integrating out the task-specific parameters to form the marginal likelihood of the data. Formally, grouping all of the data from each of the tasks as $\mathbf{X}$ and again denoting by $\mathbf{x}_{j_1}, \ldots, \mathbf{x}_{j_N}$ a sample from task $\mathcal{T}_j$, the marginal likelihood of the observed data is given by

$$p(\mathbf{X} \mid \boldsymbol{\theta}) = \prod_j \left( \int p(\mathbf{x}_{j_1}, \ldots, \mathbf{x}_{j_N} \mid \phi_j) \, p(\phi_j \mid \boldsymbol{\theta}) \, \mathrm{d}\phi_j \right) \quad . \tag{2}$$

Maximizing (2) as a function of $\boldsymbol{\theta}$ gives a point estimate for $\boldsymbol{\theta}$, an instance of a method known as empirical Bayes (Bernardo & Smith, 2006; Gelman et al., 2014) due to its use of the data to estimate the parameters of the prior distribution.

Hierarchical Bayesian models have a long history of use in both transfer learning and domain adaptation (*e.g.,* Lawrence & Platt, 2004; Yu et al., 2005; Gao et al., 2008; Daumé III, 2009; Wan et al., 2012). However, the formulation of meta-learning as hierarchical Bayes does not automatically provide an inference procedure, and furthermore, there is no guarantee that inference is tractable for expressive models with many parameters such as deep neural networks.

## 3 LINKING GRADIENT-BASED META-LEARNING & HIERARCHICAL BAYES

In this section, we connect the two independent approaches of Section 2.1 and Section 2.2 by showing that MAML can be understood as empirical Bayes in a hierarchical probabilistic model. Furthermore, we build on this understanding by showing that a choice of update rule for the task-specific parameters $\phi_j$ (*i.e.,* a choice of inner-loop optimizer) corresponds to a choice of prior over task-specific parameters, $p(\phi_j \mid \boldsymbol{\theta})$.

### 3.1 MODEL-AGNOSTIC META-LEARNING AS EMPIRICAL BAYES

In general, when performing empirical Bayes, the marginalization over task-specific parameters $\phi_j$ in (2) is not tractable to compute exactly. To avoid this issue, we can consider an approximation that makes use of a point estimate $\hat{\phi}_j$ instead of performing the integration over $\phi$ in (2). Using $\hat{\phi}_j$ as an estimator for each $\phi_j$, we may write the negative logarithm of the marginal likelihood as

$$-\log p(\mathbf{X} \mid \boldsymbol{\theta}) \approx \sum_j \left[ -\log p\left( \mathbf{x}_{j_{N+1}}, \ldots \mathbf{x}_{j_{N+M}} \mid \hat{\phi}_j \right) \right] \quad . \tag{3}$$

Setting $\hat{\phi}_j = \boldsymbol{\theta} + \alpha \nabla_{\boldsymbol{\theta}} \log p(\mathbf{x}_{j_1}, \ldots, \mathbf{x}_{j_N} \mid \boldsymbol{\theta})$ for each $j$ in (3) recovers the unscaled form of the one-step MAML objective in (1). This tells us that the MAML objective is equivalent to a maximization with respect to the meta-level parameters $\boldsymbol{\theta}$ of the marginal likelihood $p(\mathbf{X} \mid \boldsymbol{\theta})$, where a point estimate for each task-specific parameter $\phi_j$ is computed via one or a few steps of gradient descent. By taking only a few steps from the initialization at $\boldsymbol{\theta}$, the point estimate $\hat{\phi}_j$ trades off

---

**Algorithm** `MAML-HB`$(\mathscr{D})$

    Initialize $\boldsymbol{\theta}$ randomly

    **while** *not converged* **do**

        Draw $J$ samples $\mathcal{T}_1, \ldots, \mathcal{T}_J \sim p_{\mathscr{D}}(\mathcal{T})$

        Estimate $\mathbb{E}_{\mathbf{x} \sim p_{\mathcal{T}_1}(\mathbf{x})}[-\log p(\mathbf{x} \mid \boldsymbol{\theta})], \ldots, \mathbb{E}_{\mathbf{x} \sim p_{\mathcal{T}_J}(\mathbf{x})}[-\log p(\mathbf{x} \mid \boldsymbol{\theta})]$ using `ML-`$\cdots$

        Update $\boldsymbol{\theta} \leftarrow \boldsymbol{\theta} - \beta \, \nabla_{\boldsymbol{\theta}} \sum_j \mathbb{E}_{\mathbf{x} \sim p_{\mathcal{T}_j}(\mathbf{x})}[-\log p(\mathbf{x} \mid \boldsymbol{\theta})]$

    **end**

---

**Algorithm 2:** Model-agnostic meta-learning as hierarchical Bayesian inference. The choices of the subroutine `ML-`$\cdots$ that we consider are defined in Subroutine 3 and Subroutine 4.

---

**Subroutine** `ML-POINT`$(\boldsymbol{\theta}, \mathcal{T})$

    Draw $N$ samples $\mathbf{x}_1, \ldots, \mathbf{x}_N \sim p_{\mathcal{T}}(\mathbf{x})$

    Initialize $\boldsymbol{\phi} \leftarrow \boldsymbol{\theta}$

    **for** $k$ *in* $1, \ldots, K$ **do**

        Update $\boldsymbol{\phi} \leftarrow \boldsymbol{\phi} + \alpha \nabla_{\boldsymbol{\phi}} \log p(\mathbf{x}_1, \ldots, \mathbf{x}_N \mid \boldsymbol{\phi})$

    **end**

    Draw $M$ samples $\mathbf{x}_{N+1}, \ldots, \mathbf{x}_{N+M} \sim p_{\mathcal{T}}(\mathbf{x})$

    **return** $-\log p(\mathbf{x}_{N+1}, \ldots, \mathbf{x}_{N+M} \mid \boldsymbol{\phi})$

---

**Subroutine 3:** Subroutine for computing a point estimate $\hat{\boldsymbol{\phi}}$ using truncated gradient descent to approximate the marginal negative log likelihood (NLL).

minimizing the fast adaptation objective $-\log p(\mathbf{x}_{j_1}, \ldots, \mathbf{x}_{j_N} \mid \boldsymbol{\theta})$ with staying close in value to the parameter initialization $\boldsymbol{\theta}$.

We can formalize this trade-off by considering the linear regression case. Recall that the *maximum a posteriori* (MAP) estimate of $\boldsymbol{\phi}_j$ corresponds to the global mode of the posterior $p(\boldsymbol{\phi}_j \mid \mathbf{x}_{j_1}, \ldots \mathbf{x}_{j_N}, \boldsymbol{\theta}) \propto p(\mathbf{x}_{j_1}, \ldots \mathbf{x}_{j_N} \mid \boldsymbol{\phi}_j) p(\boldsymbol{\phi}_j \mid \boldsymbol{\theta})$. In the case of a linear model, early stopping of an iterative gradient descent procedure to estimate $\boldsymbol{\phi}_j$ is exactly equivalent to MAP estimation of $\boldsymbol{\phi}_j$ under the assumption of a prior that depends on the number of descent steps as well as the direction in which each step is taken. In particular, write the input examples as $\mathbf{X}$ and the vector of regression targets as $\mathbf{y}$, omit the task index from $\boldsymbol{\phi}$, and consider the gradient descent update

$$\boldsymbol{\phi}_{(k)} = \boldsymbol{\phi}_{(k-1)} - \alpha \nabla_{\boldsymbol{\phi}} \left[ \|\mathbf{y} - \mathbf{X}\boldsymbol{\phi}\|_2^2 \right]_{\boldsymbol{\phi} = \boldsymbol{\phi}_{(k-1)}}$$

$$= \boldsymbol{\phi}_{(k-1)} - \alpha \mathbf{X}^{\mathrm{T}} \left( \mathbf{X}\boldsymbol{\phi}_{(k-1)} - \mathbf{y} \right) \tag{4}$$

for iteration index $k$ and learning rate $\alpha \in \mathbb{R}^+$. Santos (1996) shows that, starting from $\boldsymbol{\phi}_{(0)} = \boldsymbol{\theta}$, $\boldsymbol{\phi}_{(k)}$ in (4) solves the regularized linear least squares problem

$$\min \left( \|\mathbf{y} - \mathbf{X}\boldsymbol{\phi}\|_2^2 + \|\boldsymbol{\theta} - \boldsymbol{\phi}\|_{\mathbf{Q}}^2 \right) \tag{5}$$

with $\mathbf{Q}$-norm defined by $\|\mathbf{z}\|_{\mathbf{Q}} = \mathbf{z}^{\mathrm{T}} \mathbf{Q}^{-1} \mathbf{z}$ for a symmetric positive definite matrix $\mathbf{Q}$ that depends on the step size $\alpha$ and iteration index $k$ as well as on the covariance structure of $\mathbf{X}$. We describe the exact form of the dependence in Section 3.2. The minimization in (5) can be expressed as a posterior maximization problem given a conditional Gaussian likelihood over $\mathbf{y}$ and a Gaussian prior over $\boldsymbol{\phi}$. The posterior takes the form

$$p(\boldsymbol{\phi} \mid \mathbf{X}, \mathbf{y}, \boldsymbol{\theta}) \propto \mathcal{N}(\mathbf{y} ; \mathbf{X}\boldsymbol{\phi}, \mathbb{I}) \, \mathcal{N}(\boldsymbol{\phi} ; \boldsymbol{\theta}, \mathbf{Q}) \ . \tag{6}$$

Since $\boldsymbol{\phi}_{(k)}$ in (4) maximizes (6), we may conclude that $k$ iterations of gradient descent in a linear regression model with squared error exactly computes the MAP estimate of $\boldsymbol{\phi}$, given a Gaussian-noised observation model and a Gaussian prior over $\boldsymbol{\phi}$ with parameters $\boldsymbol{\mu}_0 = \boldsymbol{\theta}$ and $\boldsymbol{\Sigma}_0 = \mathbf{Q}$. Therefore, in

the case of linear regression with squared error, MAML is exactly empirical Bayes using the MAP estimate as the point estimate of $\phi$.

In the nonlinear case, MAML is again equivalent to an empirical Bayes procedure to maximize the marginal likelihood that uses a point estimate for $\phi$ computed by one or a few steps of gradient descent. However, this point estimate is not necessarily the global mode of a posterior. We can instead understand the point estimate given by truncated gradient descent as the value of the mode of an implicit posterior over $\phi$ resulting from an empirical loss interpreted as a negative log-likelihood, and regularization penalties and the early stopping procedure jointly acting as priors (for similar interpretations, see Sjöberg & Ljung, 1995; Bishop, 1995; Duvenaud et al., 2016).

The exact equivalence between early stopping and a Gaussian prior on the weights in the linear case, as well as the implicit regularization to the parameter initialization the nonlinear case, tells us that every iterate of truncated gradient descent is a mode of an implicit posterior. In particular, we are not required to take the gradient descent procedure of fast adaptation that computes $\hat{\phi}$ to convergence in order to establish a connection between MAML and hierarchical Bayes. MAML can therefore be understood to approximate an expectation of the marginal negative log likelihood (NLL) for each task $\mathcal{T}_j$ as

$$\mathbb{E}_{\mathbf{x} \sim p_{\mathcal{T}_j}(\mathbf{x})} \left[ - \log p\left( \mathbf{x} \mid \boldsymbol{\theta} \right) \right] \approx \frac{1}{M} \sum_m - \log p\left( \mathbf{x}_{j_{N+m}} \mid \hat{\boldsymbol{\phi}}_j \right)$$

using the point estimate $\hat{\boldsymbol{\phi}}_j = \boldsymbol{\theta} + \alpha \, \nabla_{\boldsymbol{\theta}} \log p( \mathbf{x}_{j_n} \mid \boldsymbol{\theta} )$ for single-step fast adaptation.

The algorithm for MAML as probabilistic inference is given in Algorithm 2; Subroutine 3 computes each marginal NLL using the point estimate of $\hat{\phi}$ as just described. Formulating MAML in this way, as probabilistic inference in a hierarchical Bayesian model, motivates the interpretation in Section 3.2 of using various meta-optimization algorithms to induce a prior over task-specific parameters.

## 3.2 The Prior Over Task-Specific Parameters

From Section 3.1, we may conclude that early stopping during fast adaptation is equivalent to a specific choice of a prior over task-specific parameters, $p( \boldsymbol{\phi}_j \mid \theta )$. We can better understand the role of early stopping in defining the task-specific parameter prior in the case of a quadratic objective. Omit the task index from $\phi$ and $\mathbf{x}$, and consider a second-order approximation of the fast adaptation objective $\ell(\boldsymbol{\phi}) = - \log p( \mathbf{x}_1 \ldots, \mathbf{x}_N \mid \boldsymbol{\phi} )$ about a minimum $\boldsymbol{\phi}^*$:

$$\ell(\boldsymbol{\phi}) \approx \tilde{\ell}(\boldsymbol{\phi}) := \tfrac{1}{2} \|\boldsymbol{\phi} - \boldsymbol{\phi}^*\|^2_{\mathbf{H}^{-1}} + \ell(\boldsymbol{\phi}^*) \tag{7}$$

where the Hessian $\mathbf{H} = \nabla^2_{\boldsymbol{\phi}} \ell(\boldsymbol{\phi}^*)$ is assumed to be positive definite so that $\tilde{\ell}$ is bounded below. Furthermore, consider using a curvature matrix $\mathcal{B}$ to precondition the gradient in gradient descent, giving the update

$$\boldsymbol{\phi}_{(k)} = \boldsymbol{\phi}_{(k-1)} - \mathcal{B} \, \nabla_{\boldsymbol{\phi}} \, \tilde{\ell}(\boldsymbol{\phi}_{(k-1)}) \ . \tag{8}$$

If $\mathcal{B}$ is diagonal, we can identify (8) as a Newton method with a diagonal approximation to the inverse Hessian; using the inverse Hessian evaluated at the point $\boldsymbol{\phi}_{(k-1)}$ recovers Newton's method itself. On the other hand, meta-learning the matrix $\mathcal{B}$ matrix via gradient descent provides a method to incorporate task-general information into the covariance of the fast adaptation prior, $p( \phi \mid \theta )$. For instance, the meta-learned matrix $\mathcal{B}$ may encode correlations between parameters that dictates how such parameters are updated relative to each other.

Formally, taking $k$ steps of gradient descent from $\boldsymbol{\phi}_{(0)} = \boldsymbol{\theta}$ using the update rule in (8) gives a $\boldsymbol{\phi}_{(k)}$ that solves

$$\min \left( \|\boldsymbol{\phi} - \boldsymbol{\phi}^*\|^2_{\mathbf{H}^{-1}} + \|\boldsymbol{\phi}_{(0)} - \boldsymbol{\phi}\|^2_{\mathbf{Q}} \right) \ . \tag{9}$$

The minimization in (9) corresponds to taking a Gaussian prior $p( \phi \mid \theta )$ with mean $\boldsymbol{\theta}$ and covariance $\mathbf{Q}$ for $\mathbf{Q} = \mathbf{O} \boldsymbol{\Lambda}^{-1}((\mathbb{I} - \mathbf{B}\boldsymbol{\Lambda})^{-k} - \mathbb{I})\mathbf{O}^{\mathrm{T}}$ (Santos, 1996) where $\mathbf{B}$ is a diagonal matrix that

results from a simultaneous diagonalization of $\mathbf{H}$ and $\mathcal{B}$ as $\mathbf{O}^{\mathrm{T}}\mathbf{H}\mathbf{O} = \mathrm{diag}(\lambda_1, \ldots, \lambda_n) = \mathbf{\Lambda}$ and $\mathbf{O}^{\mathrm{T}}\mathcal{B}^{-1}\mathbf{O} = \mathrm{diag}(b_1, \ldots, b_n) = \mathbf{B}$ with $b_i, \lambda_i \geq 0$ for $i = 1, \ldots, n$ (Theorem 8.7.1 in Golub & Van Loan, 1983). If the true objective is indeed quadratic, then, assuming the data is centered, $\mathbf{H}$ is the unscaled covariance matrix of features, $\mathbf{X}^{\mathrm{T}}\mathbf{X}$.

# 4 IMPROVING MODEL-AGNOSTIC META-LEARNING

Identifying MAML as a method for probabilistic inference in a hierarchical model allows us to develop novel improvements to the algorithm. In Section 4.1, we consider an approach from Bayesian parameter estimation to improve the MAML algorithm, and in Section 4.2, we discuss how to make this procedure computationally tractable for high-dimensional models.

## 4.1 LAPLACE'S METHOD OF INTEGRATION

We have shown that the MAML algorithm is an empirical Bayes procedure that employs a point estimate for the mid-level, task-specific parameters in a hierarchical Bayesian model. However, the use of this point estimate may lead to an inaccurate point approximation of the integral in (2) if the posterior over the task-specific parameters, $p(\boldsymbol{\phi}_j \mid \mathbf{x}_{j_{N+1}}, \ldots, \mathbf{x}_{j_{N+M}}, \boldsymbol{\theta})$, is not sharply peaked at the value of the point estimate. The Laplace approximation (Laplace, 1986; MacKay, 1992b;a) is applicable in this case as it replaces a point estimate of an integral with the volume of a Gaussian centered at a mode of the integrand, thereby forming a local quadratic approximation.

We can make use of this approximation to incorporate uncertainty about the task-specific parameters into the MAML algorithm at fast adaptation time. In particular, suppose that each integrand in (2) has a mode $\boldsymbol{\phi}_j^*$ at which it is locally well-approximated by a quadratic function. The Laplace approximation uses a second-order Taylor expansion of the negative log posterior in order to approximate each integral in the product in (2) as

$$\int p\left(\mathbf{X}_j \mid \boldsymbol{\phi}_j\right) p\left(\boldsymbol{\phi}_j \mid \boldsymbol{\theta}\right) \mathrm{d}\boldsymbol{\phi}_j \approx p\left(\mathbf{X}_j \mid \boldsymbol{\phi}_j^*\right) p\left(\boldsymbol{\phi}_j^* \mid \boldsymbol{\theta}\right) \det(\mathbf{H}_j/2\pi)^{-\frac{1}{2}} \qquad (10)$$

where $\mathbf{H}_j$ is the Hessian matrix of second derivatives of the negative log posterior.

Classically, the Laplace approximation uses the MAP estimate for $\boldsymbol{\phi}_j^*$, although any mode can be used as an expansion site provided the integrand is well enough approximated there by a quadratic. We use the point estimate $\hat{\boldsymbol{\phi}}_j$ uncovered by fast adaptation, in which case the MAML objective in (1) becomes an appropriately scaled version of the approximate marginal likelihood

$$-\log p\left(\mathbf{X} \mid \boldsymbol{\theta}\right) \approx \sum_j \left[ -\log p\left(\mathbf{X}_j \mid \hat{\boldsymbol{\phi}}_j\right) - \log p\left(\hat{\boldsymbol{\phi}}_j \mid \boldsymbol{\theta}\right) + \tfrac{1}{2}\log \det(\mathbf{H}_j) \right] \quad . \qquad (11)$$

The term $\log p(\hat{\boldsymbol{\phi}}_j \mid \boldsymbol{\theta})$ results from the implicit regularization imposed by early stopping during fast adaptation, as discussed in Section 3.1. The term $^1\!/_2 \log \det(\mathbf{H}_j)$, on the other hand, results from the Laplace approximation and can be interpreted as a form of regularization that penalizes model complexity.

## 4.2 USING CURVATURE INFORMATION TO IMPROVE MAML

Using (11) as a training criterion for a neural network model is difficult due to the required computation of the determinant of the Hessian of the log posterior $\mathbf{H}_j$, which itself decomposes into a sum of the Hessian of the log likelihood and the Hessian of the log prior as

$$\mathbf{H}_j = \nabla^2_{\boldsymbol{\phi}_j} \left[ -\log p\left(\mathbf{X}_j \mid \boldsymbol{\phi}_j\right) \right] + \nabla^2_{\boldsymbol{\phi}_j} \left[ -\log p\left(\boldsymbol{\phi}_j \mid \boldsymbol{\theta}\right) \right] \quad .$$

In our case of early stopping as regularization, the prior over task-specific parameters $p(\boldsymbol{\phi}_j \mid \boldsymbol{\theta})$ is implicit and thus no closed form is available for a general model. Although we may use the quadratic approximation derived in Section 3.2 to obtain an approximate Gaussian prior, this prior is not diagonal and does not, to our knowledge, have a convenient factorization. Therefore, in our experiments, we instead use a simple approximation in which the prior is approximated as a diagonal Gaussian with precision $\tau$. We keep $\tau$ fixed, although this parameter may be cross-validated for improved performance.

---

**Subroutine** `ML-LAPLACE` $(\boldsymbol{\theta}, \mathcal{T})$
    Draw $N$ samples $\mathbf{x}_1, \ldots, \mathbf{x}_N \sim p_{\mathcal{T}}(\mathbf{x})$
    Initialize $\boldsymbol{\phi} \leftarrow \boldsymbol{\theta}$
    **for** $k$ *in* $1, \ldots, K$ **do**
        | Update $\boldsymbol{\phi} \leftarrow \boldsymbol{\phi} + \alpha \nabla_{\boldsymbol{\phi}} \log p(\mathbf{x}_1, \ldots, \mathbf{x}_N \mid \boldsymbol{\phi})$
    **end**
    Draw $M$ samples $\mathbf{x}_{N+1}, \ldots, \mathbf{x}_{N+M} \sim p_{\mathcal{T}}(\mathbf{x})$
    Estimate quadratic curvature $\hat{\mathbf{H}}$
    **return** $-\log p(\mathbf{x}_{N+1}, \ldots, \mathbf{x}_{N+M} \mid \boldsymbol{\phi}) + \eta \log \det(\hat{\mathbf{H}})$

---

**Subroutine 4:** Subroutine for computing a Laplace approximation of the marginal likelihood.

Similarly, the Hessian of the log likelihood is intractable to form exactly for all but the smallest models, and furthermore, is not guaranteed to be positive definite at all points, possibly rendering the Laplace approximation undefined. To combat this, we instead seek a curvature matrix $\hat{\mathbf{H}}$ that approximates the quadratic curvature of a neural network objective function. Since it is well-known that the curvature associated with neural network objective functions is highly non-diagonal (*e.g.,* Martens, 2016), a further requirement is that the matrix have off-diagonal terms.

Due to the difficulties listed above, we turn to second order gradient descent methods, which precondition the gradient with an inverse curvature matrix at each iteration of descent. The Fisher information matrix (Fisher, 1925) has been extensively used as an approximation of curvature, giving rise to a method known as natural gradient descent (Amari, 1998). A neural network with an appropriate choice of loss function is a probabilistic model and therefore defines a Fisher information matrix. Furthermore, the Fisher information matrix can be seen to define a convex quadratic approximation to the objective function of a probabilistic neural model (Pascanu & Bengio, 2014; Martens, 2014). Importantly for our use case, the Fisher information matrix is positive definite by definition as well as non-diagonal.

However, the Fisher information matrix is still expensive to work with. Martens & Grosse (2015) developed Kronecker-factored approximate curvature (K-FAC), a scheme for approximating the curvature of the objective function of a neural network with a block-diagonal approximation to the Fisher information matrix. Each block corresponds to a unique layer in the network, and each block is further approximated as a Kronecker product (see Van Loan, 2000) of two much smaller matrices by assuming that the second-order statistics of the input activation and the back-propagated derivatives within a layer are independent. These two approximations ensure that the inverse of the Fisher information matrix can be computed efficiently for the natural gradient.

For the Laplace approximation, we are interested in the determinant of a curvature matrix instead of its inverse. However, we may also make use of the approximations to the Fisher information matrix from K-FAC as well as properties of the Kronecker product. In particular, we use the fact that the determinant of a Kronecker product is the product of the exponentiated determinants of each of the factors, and that the determinant of a block diagonal matrix is the product of the determinants of the blocks (Van Loan, 2000). The determinants for each factor can be computed as efficiently as the inverses required by K-FAC, in $\mathcal{O}(d^3)$ time for a $d$-dimensional Kronecker factor.

We make use of the Laplace approximation and K-FAC to replace Subroutine 3, which computes the task-specific marginal NLLs using a point estimate for $\hat{\boldsymbol{\phi}}$. We call this method the Lightweight Laplace Approximation for Meta-Adaptation (LLAMA), and give a replacement subroutine in Subroutine 4.

## 5    EXPERIMENTAL EVALUATION

The goal of our experiments is to evaluate if we can use our probabilistic interpretation of MAML to generate samples from the distribution over adapted parameters, and futhermore, if our method can be applied to large-scale meta-learning problems such as *mini*ImageNet.

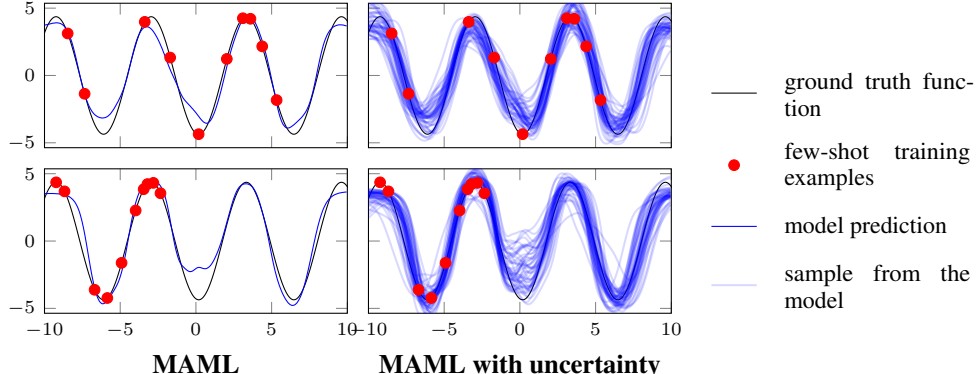

**Figure 5:** Our method is able to meta-learn a model that can quickly adapt to sinusoids with varying phases and amplitudes, and the interpretation of the method as hierarchical Bayes makes it practical to directly sample models from the posterior. In this figure, we illustrate various samples from the posterior of a model that is meta-trained on different sinusoids, when presented with a few datapoints (in red) from a new, previously unseen sinusoid. Note that the random samples from the posterior predictive describe a distribution of functions that are all sinusoidal and that there is increased uncertainty when the datapoints are less informative (*i.e.,* when the datapoints are sampled only from the lower part of the range input, shown in the bottom-right example).

## 5.1 WARMUP: TOY NONLINEAR MODEL

The connection between MAML and hierarchical Bayes suggests that we should expect MAML to behave like an algorithm that learns the mean of a Gaussian prior on model parameters, and uses the mean of this prior as an initialization during fast adaptation. Using the Laplace approximation to the integration over task-specific parameters as in (10) assumes a task-specific parameter posterior with mean at the adapted parameters $\hat{\phi}$ and covariance equal to the inverse Hessian of the log posterior evaluated at the adapted parameter value. Instead of simply using this density in the Laplace approximation as an additional regularization term as in (11), we may sample parameters $\phi_j$ from this density and use each set of sampled parameters to form a set of predictions for a given task.

To illustrate this relationship between MAML and hierarchical Bayes, we present a meta-dataset of sinusoid tasks in which each task involves regressing to the output of a sinusoid wave in Figure 5. Variation between tasks is obtained by sampling the amplitude uniformly from $[0.1, 5.0]$ and the phase from $[0, \pi]$. During training and for each task, 10 input datapoints are sampled uniformly from $[-10.0, 10.0]$ and the loss is the mean squared error between the prediction and the true value.

We observe in Figure 5 that our method allows us to directly sample models from the task-specific parameter distribution after being presented with 10 datapoints from a new, previously unseen sinusoid curve. In particular, the column on the right of Figure 5 demonstrates that the sampled models display an appropriate level of uncertainty when the datapoints are ambiguous (as in the bottom right).

## 5.2 LARGE-SCALE EXPERIMENT: *mini*IMAGENET

We evaluate LLAMA on the miniImageNet Ravi & Larochelle (2017) 1-shot, 5-way classification task, a standard benchmark in few-shot classification. *mini*ImageNet comprises 64 training classes, 12 validation classes, and 24 test classes. Following the setup of Vinyals et al. (2016), we structure the $N$-shot, $J$-way classification task as follows: The model observes $N$ instances of $J$ unseen classes, and is evaluated on its ability to classify $M$ new instances within the $J$ classes.

We use a neural network architecture standard to few-shot classification (*e.g.,* Vinyals et al., 2016; Ravi & Larochelle, 2017), consisting of 4 layers with $3 \times 3$ convolutions and $64$ filters, followed by batch normalization (BN) (Ioffe & Szegedy, 2015), a ReLU nonlinearity, and $2 \times 2$ max-pooling. For the scaling variable $\beta$ and centering variable $\gamma$ of BN (see Ioffe & Szegedy, 2015), we ignore the fast adaptation update as well as the Fisher factors for K-FAC. We use Adam (Kingma & Ba, 2014) as the meta-optimizer, and standard batch gradient descent with a fixed learning rate to update the model

| Model | 5-way acc. (%) 1-shot | | |
|---|---|---|---|
| **Fine-tuning**[*] | 28.86 | $\pm$ | 0.54 |
| **Nearest Neighbor**[*] | 41.08 | $\pm$ | 0.70 |
| **Matching Networks FCE** (Vinyals et al., 2016)[*] | 43.56 | $\pm$ | 0.84 |
| **Meta-Learner LSTM** (Ravi & Larochelle, 2017)[*] | 43.44 | $\pm$ | 0.77 |
| **SNAIL** (Mishra et al., 2018)[**] | 45.1 | $\pm$ | —— |
| **Prototypical Networks** (Snell et al., 2017)[***] | 46.61 | $\pm$ | 0.78 |
| **mAP-DLM** (Triantafillou et al., 2017) | 49.82 | $\pm$ | 0.78 |
| **MAML** (Finn et al., 2017) | 48.70 | $\pm$ | 1.84 |
| **LLAMA (Ours)** | 49.40 | $\pm$ | 1.83 |

**Table 1:** One-shot classification performance on the *mini*ImageNet test set, with comparison methods ordered by one-shot performance. All results are averaged over 600 test episodes, and we report $95\%$ confidence intervals. [*]Results reported by Ravi & Larochelle (2017). [**]We report test accuracy for a comparable architecture.[1][***]We report test accuracy for models matching train and test "shot" and "way".

during fast adaptation. LLAMA requires the prior precision term $\tau$ as well as an additional parameter $\eta \in \mathbb{R}^+$ that weights the regularization term $\log \det \hat{\mathbf{H}}$ contributed by the Laplace approximation. We fix $\tau = 0.001$ and selected $\eta = 10^{-6}$ via cross-validation; all other parameters are set to the values reported in Finn et al. (2017).

We find that LLAMA is practical enough to be applied to this larger-scale problem. In particular, our TensorFlow implementation of LLAMA trains for 60,000 iterations on one TITAN Xp GPU in 9 hours, compared to 5 hours to train MAML. As shown in Table 1, LLAMA achieves comparable performance to the state-of-the-art meta-learning method by Triantafillou et al. (2017). While the gap between MAML and LLAMA is small, the improvement from the Laplace approximation suggests that a more accurate approximation to the marginalization over task-specific parameters will lead to further improvements.

## 6 RELATED WORK

Meta-learning and few-shot learning have a long history in hierarchical Bayesian modeling (*e.g.,* Tenenbaum, 1999; Fei-Fei et al., 2003; Lawrence & Platt, 2004; Yu et al., 2005; Gao et al., 2008; Daumé III, 2009; Wan et al., 2012). A related subfield is that of transfer learning, which has used hierarchical Bayes extensively (*e.g.,* Raina et al., 2006). A variety of inference methods have been used in Bayesian models, including exact inference (Lake et al., 2011), sampling methods (Salakhutdinov et al., 2012), and variational methods (Edwards & Storkey, 2017). While some prior works on hierarchical Bayesian models have proposed to handle basic image recognition tasks, the complexity of these tasks does not yet approach the kinds of complex image recognition problems that can be solved by discriminatively trained deep networks, such as the *mini*ImageNet experiment in our evaluation (Mansinghka et al., 2013).

Recently, the Omniglot benchmark Lake et al. (2016) has rekindled interest in the problem of learning from few examples. Modern methods accomplish few-shot learning either through the design of network architectures that ingest the few-shot training samples directly (*e.g.,* Koch, 2015; Vinyals et al., 2016; Snell et al., 2017; Hariharan & Girshick, 2017; Triantafillou et al., 2017), or formulating the problem as one of *learning to learn*, or *meta-learning* (*e.g.,* Schmidhuber, 1987; Bengio et al., 1991; Schmidhuber, 1992; Bengio et al., 1992). A variety of inference methods have been used in Bayesian models, including exact inference (Lake et al., 2011), sampling methods (Salakhutdinov et al., 2013), and variational methods (Edwards & Storkey, 2017).

Our work bridges the gap between gradient-based meta-learning methods and hierarchical Bayesian modeling. Our contribution is not to formulate the meta-learning problem as a hierarchical Bayesian

---

[1]Improved performance on *mini*ImageNet has been reported by several works (Mishra et al., 2017; Munkhdalai & Yu, 2017; Sung et al., 2017) by making use of a model architecture with significantly more parameters than the methods in Table 1. Since we do not explore variations in neural network architecture in this work, we omit such results from the table.

model, but instead to formulate a gradient-based meta-learner as hierarchical Bayesian inference, thus providing a way to efficiently perform posterior inference in a model-agnostic manner.

## 7    CONCLUSION

We have shown that model-agnostic meta-learning (MAML) estimates the parameters of a prior in a hierarchical Bayesian model. By casting gradient-based meta-learning within a Bayesian framework, our analysis opens the door to novel improvements inspired by probabilistic machinery.

As a step in this direction, we propose an extension to MAML that employs a Laplace approximation to the posterior distribution over task-specific parameters. This technique provides a more accurate estimate of the integral that, in the original MAML algorithm, is approximated via a point estimate. We show how to estimate the quantity required by the Laplace approximation using Kronecker-factored approximate curvature (K-FAC), a method recently proposed to approximate the quadratic curvature of a neural network objective for the purpose of a second-order gradient descent technique.

Our contribution illuminates the road to exploring further connections between gradient-based meta-learning methods and hierarchical Bayesian modeling. For instance, in this work we assume that the predictive distribution over new data-points is narrow and well-approximated by a point estimate. We may instead employ methods that make use of the variance of the distribution over task-specific parameters in order to model the predictive density over examples from a novel task.

Furthermore, it is known that the Laplace approximation is inaccurate in cases where the integral is highly skewed, or is not unimodal and thus is not amenable to approximation by a single Gaussian mode. This could be solved by using a finite mixture of Gaussians, which can approximate many density functions arbitrarily well (Sorenson & Alspach, 1971; Alspach & Sorenson, 1972). The exploration of additional improvements such as this is an exciting line of future work.

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
