# OpenReview forum: "Recasting Gradient-Based Meta-Learning as Hierarchical Bayes"
_ICLR.cc/2018/Conference — Accept (Poster)_

### Official Review · AnonReviewer1 · 2017-11-27

**Rating:** 6
**Confidence:** 3

**Review:**

Summary
The paper presents an interesting view on the recently proposed MAML formulation of meta-learning (Finn et al). The main contribution is a) insight into the connection between the MAML procedure and MAP estimation in an equivalent linear hierarchical Bayes model with explicit priors, b) insight into the connection between MAML and MAP estimation in non-linear HB models with implicit priors, c) based on these insights, the paper proposes a variant of MALM using a Laplace approximation (with additional approximations for the covariance matrix. The paper finally provides an evaluation on the mini ImageNet problem without significantly improving on the MAML results on the same task.

Pro:
-            The topic is timely and of relevance to the ICLR community continuing a current trend in building meta-learning system for few-shot learning.
-            Provides valuable insight into the MAML objective and its relation to probabilistic models

Con:
-            The paper is generally well-written but I find (as a non-meta-learner expert) that certain fundamental aspects could have been explained better or in more detail (see below for details).
-            The toy example is quite difficult to interpret the first time around and does not provide any empirical insight into the converge of the proposed method (compared to e.g. MAML)
-            I do not think the empirical results provide enough evidence that it is a useful/robust method. Especially it does not provide insight into which types of problems (small/large, linear/ non-linear) the method is applicable to.


Detailed comments/questions:
-            The use of Laplace approximation is (in the paper) motivated from a probabilistic/Bayes and uncertainty point-of-view. It would, however, seem that the truncated iterations do not result in the approximation being very accurate during optimization as the truncation does not result in the approximation being created at a mode. Could the authors perhaps comment on:
a) whether it is even meaningful to talk about the approximations as probabilistic distribution during the optimization (given the psd approximation to the Hessian), or does it only make sense after convergence?
b) the consequence of the approximation errors on the general convergence of the proposed method (consistency and rate)

-            Sec 4.1, p5: Last equation: Perhaps useful to explain the term $log(\phi_j^* | \theta)$ and why it is not in subroutine 4 . Should $\phi^*$  be $\hat \phi$ ?
-            Sec 4.2: “A straightforward…”: I think it would improve readability to refer back to the to the previous equation (i.e. H) such that it is clear what is meant by “straightforward”.
-            Sec 4.2: Several ideas are being discussed in Sec 4.2 and it is not entirely clear to me what has actually been adopted here; perhaps consider formalizing the actual computations in Subroutine 4 – and provide a clearer argument (preferably proof) that this leads to consistent and robust estimator of \theta.
-            It is not clear from the text or experiment how the learning parameters are set.
-            Sec 5.1: It took some effort to understand exactly what was going on in the example and particular figure 5.1; e.g., in the model definition in the body text there is no mention of the NN mentioned/used in figure 5, the blue points are not defined in the caption, the terminology e.g.  “pre-update density” is new at this point. I think it would benefit the readability to provide the reader with a bit more guidance.
-            Sec 5.1: While the qualitative example is useful (with a bit more text), I believe it would have been more convincing with a quantitative example to demonstrate e.g. the convergence of the proposal compared to std MAML and possibly compare to a std Bayesian inference method from the HB formulation of the problem (in the linear case)
-            Sec 5.2: The abstract clams increased performance over MAML but the empirical results do not seem to be significantly better than MAML ? I find it quite difficult to support the specific claim in the abstract from the results without adding a comment about the significance.
-            Sec 5.2: The authors have left out “Mishral et al” from the comparison due to the model being significantly larger than others. Could the authors provide insight into why they did not use the ResNet structure from the  tcml paper in their L-MLMA scheme ?
-            Sec 6+7: The paper clearly states that it is not the aim to (generally) formulate the MAML as a HB. Given the advancement in gradient based inference for HB the last couple of years (e.g. variational, nested laplace , expectation propagation etc) for explicit models, could the authors perhaps indicate why they believe their approach of looking directly to the MAML objective is more scalable/useful than trying to formulate the same or similar objective in an explicit HB model and using established inference methods from that area ?

Minor:
-            Sec 4.1 “…each integral in the sum in (2)…” eq 2 is a product

---

> ### Author Response · Authors · 2018-01-06
> **Revised paper, addressing concerns of Reviewer 1 [1/2]**
>
> We thank R1 for thorough and constructive comments! We have attempted to address all concerns to the best of our ability.
>
> > “The toy example is quite difficult to interpret the first time around and does not provide any empirical insight into the converge of the proposed method (compared to e.g. MAML)…I do not think the empirical results provide enough evidence that it is a useful/robust method. Especially it does not provide insight into which types of problems (small/large, linear/ non-linear) the method is applicable to. ”
>
> We have substantially revised the toy example in Figure 5 and its explanation in the text in Section 5.1 to better demonstrate the proposed novel algorithm. In Figure 5, we show various samples from the posterior of a model that is meta-trained on different sinusoids, when presented with a few datapoints (in red) from a new, previously unseen sinusoid. This sampling procedure is motivated by the connection we have made between MAML and HB inference. We emphasize that the quantified uncertainty evident in Figure 5 is indeed a desirable quality in a model that learns from a small amount of data.
>
> > “The use of Laplace approximation is (in the paper) motivated from a probabilistic/Bayes and uncertainty point-of-view. It would, however, seem that the truncated iterations do not result in the approximation being very accurate during optimization as the truncation does not result in the approximation being created at a mode.
> > Could the authors perhaps comment on:
> > a) whether it is even meaningful to talk about the approximations as probabilistic distribution during the optimization (given the psd approximation to the Hessian), or does it only make sense after convergence?
> > b) the consequence of the approximation errors on the general convergence of the proposed method (consistency and rate)”
>
> We have revised the paper in Section 3.1 to better convey the following: The exact equivalence between early stopping and a Gaussian prior on the weights in the linear case, as well as the implicit regularization to the parameter initialization in the nonlinear case, tells us that *every iterate of truncated gradient descent is a mode of an implicit posterior.* Therefore, in making this approximation, we are not required to take the gradient descent procedure of fast adaptation to convergence. We thus emphasize that the PSD approximation to the curvature provided by KFAC is indeed justifiable even before convergence.
>
> > Sec 4.2: Several ideas are being discussed in Sec 4.2 and it is not entirely clear to me what has actually been adopted here; perhaps consider formalizing the actual computations in Subroutine 4 – and provide a clearer argument (preferably proof) that this leads to consistent and robust estimator of \theta.
>
> Regarding the justification of using KFAC and the Laplace approximation to estimate \theta: We employ the insight from Martens (2014). In summary, for the Laplace approximation, we require a curvature matrix that would ideally be the Hessian. However, it is infeasible to compute the Hessian for all but the simplest models. In its place, we use the KFAC approximation to the Fisher; the Fisher itself can be seen as an approximation to the Hessian as: 1) It corresponds to the expected Hessian under the model's own predictive distribution; and 2) It is equivalent under common loss functions (such as cross-entropy and squared error, which we employ) to the Generalized Gauss Newton (GGN) Matrix (Pascanu & Bengio 2014, Martens 2014). We would note that we are not the first to use the GGN as an approximation to the Hessian (see, for example, Martens 2010, Vinyals & Popev, 2012).
>
> Regarding the computation of the Laplacian loss in practice: In subroutine 4, we replace H-hat with the approximation to the Fisher found in Eq. (2) of Ba et al. (2017).
>
> > It is not clear from the text or experiment how the learning parameters are set.
>
> We have clarified this in Section 5.2:  We chose the regularization weight of 10^-6 via cross-validation; all other parameters are set to the values reported in Finn et al. (2017).
>
> > Comments re: Section 5.1
>
> We apologize for the unclear diagram in the previous version of the paper. The toy example has been substantially revised in Figure 5 and elaborated in Section 5.1, as per our comment above.
>
> >  Sec 5.2: The abstract clams increased performance over MAML but the empirical results do not seem to be significantly better than MAML ?
>
> We note that Triantafillou et al. (2017) in NIPS 2017 reported a similar improvement after MAML was published in ICML 2017, and so the standard seems to be that an improvement of about 1% is publishable.

---

> ### Author Response · Authors · 2018-01-06
> **Revised paper, addressing concerns of Reviewer 1 [2/2]**
>
> > “Sec 5.2: The authors have left out “Mishral et al” from the comparison due to the model being significantly larger than others. Could the authors provide insight into why they did not use the ResNet structure from the  tcml paper in their L-MLMA scheme ?”
>
> Please see our detailed discussion with an author of TCML in the OpenReview comment thread (https://openreview.net/forum?id=BJ_UL-k0b&noteId=r1aR9l5lG). In summary, our contribution is to reinterpret MAML as approximate inference in a hierarchical Bayesian model, rather than to provide an exhaustive empirical comparison over neural network architectures (as the choice of architecture is largely orthogonal to the training loss or algorithm). Furthermore, the majority of other prior few-shot learning methods used the smaller architecture, so we felt that standardizing the architecture would provide a more informative comparison. Since we were able to obtain a number for SNAIL/TCML using the same architecture, we believe that this adequately rounds out the comparisons.
>
> > “Sec 6+7: The paper clearly states that it is not the aim to (generally) formulate the MAML as a HB. Given the advancement in gradient based inference for HB the last couple of years (e.g. variational, nested laplace , expectation propagation etc) for explicit models, could the authors perhaps indicate why they believe their approach of looking directly to the MAML objective is more scalable/useful than trying to formulate the same or similar objective in an explicit HB model and using established inference methods from that area ?”
>
> We intend the connection between MAML and HB to provide an avenue to incorporate insights from gradient-based inference, not as an explicit alternative to established inference procedures. To clarify, with the Laplace approximation, we are making the assumption that the posterior over \phi is a unimodal Gaussian with mean centered at the point estimate computed by a few steps of gradient descent during fast adaptation, and with covariance equal to the inverse Hessian evaluated at that point. However, we are not restricted to this assumption — we could potentially use another inference method (such as the nested Laplace approximation, variational Bayes, expectation propagation, or Hamiltonian Monte Carlo) to compute a more complex posterior distribution over \phi and to potentially improve performance. We may also incorporate insights from the recent literature on interpreting gradient methods as forms of probabilistic inference (e.g., Zhang & Sun et al. 2017) due the gradient-based nature of our method. This is interesting future work!
>
> If the reviewer has specific suggestions for related works that present generic inference methods (gradient-based or otherwise) that can reliably deal with high-dimensional stimuli and high-dimensional models (especially those that deal with raw images), we would be grateful to hear of them.
>
> > Sec 4.1 “…each integral in the sum in (2)…” eq 2 is a product
>
> Fixed — thank you!
>
>
> We encourage R2 to let us know of any additional questions or concerns about the clarity of the paper (especially things that could make the work clearer to a non-meta-learning audience).
>
>
> =========================================
> References
>
> Ba (2017). “Distributed Second-Order Optimization using Kronecker-Factored Approximations.” In ICLR 2017.
> Martens (2010). "Deep learning via Hessian-free optimization." In ICML 2010 http://www.cs.toronto.edu/~jmartens/docs/Deep_HessianFree.pdf
> Martens (2014). "New insights and perspectives on the natural gradient method." arXiv preprint arXiv:1412.1193. https://arxiv.org/abs/1412.1193
> Pascanu & Bengio (2013). "Revisiting natural gradient for deep networks." arXiv preprint arXiv:1301.3584. https://arxiv.org/abs/1301.3584
> Triantafillou et al. (2017). “Few-Shot Learning Through an Information Retrieval Lens.” In NIPS 2017. https://arxiv.org/abs/1707.02610
> Vinyals & Povey (2012). “Krylov Subspace Descent for Deep Learning.” In AISTATS 2012. https://arxiv.org/abs/1111.4259
> Zhang & Sun et al. (2017). “Noisy Natural Gradient as Variational Inference.” https://arxiv.org/abs/1712.02390

---

### Official Review · AnonReviewer2 · 2017-11-27
**Novel view on MAML, well presented.**

**Rating:** 7
**Confidence:** 3

**Review:**

The paper reformulates the model-agnostic meta-learning algorithm (MAML) in terms of inference for parameters of a prior distribution in a hierarchical Bayesian model. This provides an interesting and, as far as I can tell, novel view on MAML. The paper uses this view to improve the MAML algorithm. The writing of the paper is excellent. Experimental evalution is well done against a number of recently developed alternative methods in favor of the presented method, except for TCML which has been exluded using a not so convincing argument. The overview of the literature is also very well done.

---

> ### Author Response · Authors · 2018-01-06
> **Addressing comment of Reviewer 2**
>
> We thank R2 for feedback. Regarding R2’s comment on the exclusion of TCML from the miniImageNet results table: Our detailed discussion with an author of TCML is in the OpenReview comment thread (https://openreview.net/forum?id=BJ_UL-k0b&noteId=r1aR9l5lG). In summary, our contribution is to reinterpret MAML as approximate inference in a hierarchical Bayesian model, rather than to provide an exhaustive empirical comparison over neural network architectures (as the choice of architecture is largely orthogonal to the training loss or algorithm). Furthermore, the majority of other prior few-shot learning methods used the smaller architecture, so we felt that standardizing the architecture would provide a more informative comparison. Since we were able to obtain a number for SNAIL/TCML using the same architecture, we believe that this adequately rounds out the comparisons.

---

### Official Review · AnonReviewer3 · 2017-11-28
**Non-trivial hierarchical Bayes interpretation of MAML**

**Rating:** 7
**Confidence:** 3

**Review:**

MAML (Finn+ 2017) is recast as a hierarchical Bayesian learning procedure. In particular the inner (task) training is initially cast as point-wise max likelihood estimation, and then (sec4) improved upon by making use of the Laplace approximation. Experimental evidence of the relevance of the method is provided on a toy task involving a NIW prior of Gaussians, and the (benchmark) MiniImageNet task.

Casting MAML as HB seems a good idea. The paper does a good job of explaining the connection, but I think the presentation could be clarified. The role of the task prior and how it emerges from early stopping (ie a finite number of gradient descent steps) (sec 3.2) is original and technically non-trivial, and is a contribution of this paper.
The synthetic data experiment sec5.1 and fig5 is clearly explained and serves to additionally clarify the proposed method.
Regarding the MiniImageNet experiments, I read the exchange on TCML and agree with the authors of the paper under review. However, I recommend including the references to Mukhdalai 2017 and Sung 2017 in the footnote on TCML to strengthen the point more generically, and show that not just TCML but other non-shallow architectures are not considered for comparison here. In addition, the point made by the TCML authors is fair ("nothing prevented you from...") and I would also recommend mentioning the reviewed paper's authors' decision (not to test deeper architectures) in the footnote. This decision is in order but needs to be stated in order for the reader to form a balanced view of methods at her disposal.
The experimental performance reported Table 1 remains small and largely within one standard deviation of competitor methods.

I am assessing this paper as "7" because despite the merit of the paper, the relevance of the reformulation of MAML, and the technical steps involved in the reformulation, the paper does not eg address other forms (than L-MAML) of the task-specific subroutine ML-..., and the benchmark improvements are quite small. I think the approach is good and fruitful.


# Suggestions on readability

* I have the feeling the paper inverts $\alpha, \beta$ from their use in Finn 2017 (step size for meta- vs task-training). This is unfortunate and will certainly confuse readers; I advise carefully changing this throughout the entire paper (eg Algo 2,3,4, eq 1, last eq in sec3.1, eq in text below eq3, etc)

* I advise avoiding the use of the symbol f, which appears in only two places in Algo 2 and the end of sec 3.1. This is in part because f is given another meaning in Finn 2017, but also out of general parsimony in symbol use. (could leave the output of ML-... implicit by writing ML-...(\theta, T)_j in the $sum_j$; if absolutely needed, use another symbol than f)

* Maybe sec3 can be clarified in its structure by re-ordering points on the quadratic error function and early stopping (eg avoiding to split them between end of 3.1 and 3.2).

* sec6 "Machine learning and deep learning": I would definitely avoid this formulation, seems to tail in with all the media nonsense on "what's the difference between ML and DL ?". In addition the formulation seems to contrast ML with hierarchical Bayesian modeling, which does not make sense/ is wrong and confusing.

# Typos

* sec1 second parag: did you really mean "in the architecture or loss function"? unclear.
* sec2: over a family
* "common structure, so that" (not such that)
* orthgonal
* sec2.1 suggestion: clarify that \theta and \phi are in the same space
* sec2.2 suggestion: task-specific parameter $\phi_j$ is distinct from ... parameters $\phi_{j'}, j' \neq j}
* "unless an approximate ... is provided" (the use of the subjunctive here is definitely dated :-) )
* sec3.1 task-specific parameters $\phi_j$ (I would avoid writing just \phi altogether to distinguish in usage from \theta)
* Gaussian-noised
* approximation of the it objective
* before eq9: "that solves": well, it doesn't really "solve" the minimisation, in that it is not a minimum; reformulate this?
* sec4.1 innaccurate
* well approximated
* sec4.2 an curvature
* (Amari 1989)
* For the the Laplace
* O(n^3) : what is n ?
* sec5.2 (Ravi and L 2017)
* for the the

---

> ### Author Response · Authors · 2018-01-06
> **Revised paper, addressing comments of Reviewer 3**
>
> We thank R3 for thorough and constructive comments! We have attempted to address them to the best of our ability.
>
> We agree with R3’s characterization of the paper, but would like to clarify a small point for completeness:
>
> > “In particular the inner (task) training is initially cast as point-wise max likelihood estimation…”
>
> We cast the task-specific training in the inner loop as maximum a posteriori estimation (instead of max likelihood), in which the induced prior is a result of gradient descent with early stopping (termed “fast adaptation”). In particular, the induced prior serves to regularize the task-specific parameters to initial conditions (the parameter initialization).
>
> > “Regarding the MiniImageNet experiments…”
>
> Our detailed discussion with an author of TCML is in the OpenReview comment thread (https://openreview.net/forum?id=BJ_UL-k0b&noteId=r1aR9l5lG). In summary, our contribution is to reinterpret MAML as approximate inference in a hierarchical Bayesian model, rather than to provide an exhaustive empirical comparison over neural network architectures (as the choice of architecture is largely orthogonal to the training loss or algorithm). Furthermore, the majority of other prior few-shot learning methods used the smaller architecture, so we felt that standardizing the architecture would provide a more informative comparison. Since we were able to obtain a number for SNAIL/TCML using the same architecture, we believe that this adequately rounds out the comparisons.
>
> > The experimental performance reported Table 1 remains small and largely within one standard deviation of competitor methods.
>
> We note that Triantafillou et al. (2017) in NIPS 2017 reported a similar improvement after MAML was published in ICML 2017, and so the standard seems to be that an improvement of about 1% is publishable.
>
> > before eq9: "that solves": well, it doesn't really "solve" the minimisation, in that it is not a minimum; reformulate this?
>
> In the linear regression case, the iterate indeed solves the *regularized* minimization problem (in particular, it is a solution that obtains the best trade-off (wrt the regularization parameter) between minimal objective and regularization costs). However, the iterate indeed does not solve the *unregularized* problem.
>
> > “…the paper does not eg address other forms (than L-MAML) of the task-specific subroutine ML-...,”
>
> We could potentially use another inference method (such as the nested Laplace approximation, variational Bayes, expectation propagation, or Hamiltonian Monte Carlo) to compute a more complex posterior distribution over task-specific parameters \phi. This is an interesting extension that we leave to future work.
>
> > “# Suggestions on readability” & “# Typos”
>
> Many thanks for catching all of these corrigenda — we’ve corrected them in the revised paper (as follows for the more major points):
>
> - \alpha, \beta → \beta, \alpha
> - replaced “f” with “E_{x from task} [-\log p(x | \theta)]
> - We kept the split of early stopping & the quadratic function between 3.1, 3.2 since 3.2 is “additional material” and 3.1 is already dense. But, thank you for the suggestion.
> - reformulated related work
> - clarified that \theta and \phi are in the same space
> - O(n^3) → O(d^3) for d-dimensional Kronecker factor

---

> > ### Comment · AnonReviewer3 · 2018-01-09
> > **Response to author feedback**
> >
> > Thank you for your clarifications.
> >
> > > miniImageNet: I maintain the suggestions put forward in my review. I note that you cite Munkhdalai 2017 in your latest draft https://openreview.net/references/pdf?id=Bye6mda7z
> >
> > > performance: you write "We note that Triantafillou et al. (2017) in NIPS 2017 reported a similar improvement after MAML was published in ICML 2017, and so the standard seems to be that an improvement of about 1% is publishable."
> > I definitely disagree with the argument, which I think is specious. Certainly the absolute value of a performance improvement needs to be considered in realation with the standard deviation on the task. In addition, the fact that a paper with a weak performance improvement was published does not create a "judicial precedent" that would validate any further weak improvement as significant.

---

> > > ### Author Response · Authors · 2018-01-12
> > > **Added footnote clarifying comparison methods**
> > >
> > > Thank you for clarifying your request regarding the footnote on deeper comparison methods; we have modified the draft to include this footnote and relocated the Munkhdalai (2017) reference there.

---

### Public Comment · ~Mostafa_Rohaninejad1 · 2017-11-27
**Comparison with Prior Work**

Dear Authors,

We much appreciate the contributions made in this paper but would like to point out an issue with the writing / reporting of experiments.  In particular, with respect to the reporting on mini-ImageNet results, we wish to draw your attention to the omission of TCML from the results table (https://arxiv.org/abs/1707.03141).  To the best of our knowledge, TCML is in fact the current SOTA on this benchmark and seems important to be included to give the reader a complete picture.

A footnote says “We omit TCML (Mishra et al., 2017) as their ResNet architecture has significantly more parameters than the other methods and is thus not comparable. Their 1-shot performance is 55.71 ± 0.99”.  We do not believe this is a judicious exclusion.  The ability of TCML to work well with a larger architecture is an indication of its ability to extract signal in a more difficult optimization setting, an important characteristic for meta-learning algorithms.  Nothing is preventing other work (including yours) from using more expressive architectures.  Or if there are underlying limitations that makes such methods limited to smaller models (such as computational complexity or overfitting), this is highly relevant to the study of meta-learning, rather than something to simply be omitted.

Note we personally did test the larger models with not only TCML but also with MAML.  Our finding was that MAML was not able to benefit from the larger models, and it ended up overfitting and in fact doing worse than MAML with smaller models.

Sincerely,

Mostafa, Nikhil, Peter, and Pieter (authors of TCML)

---

> ### Author Response · Authors · 2017-11-27
> **Regarding architectural variation in the evaluation of meta-learning algorithms**
>
> We thank the authors of TCML for their comment regarding a comparison to TCML.
>
> We emphasize that the primary focus of our work is not to perform an exhaustive exploration of neural network architectures for few-shot classification, but instead to reinterpret and propose improvements to the MAML algorithm from a probabilistic perspective. This is the reason why we have chosen to work with the model architecture that, to the best of our knowledge, all prior work in this area that reports on the miniImageNet task (see below for a list) uses with the exceptions of TCML, MetaNetworks and Relation Networks. We thus consider the exploration of more expressive architectures not an omission but a standardization choice.
>
> Certainly, there is likely to be a better architecture for MAML and L-MAML, just as there is almost certainly a better architecture for matching networks, prototypical networks, and the various other meta-learning methods. But the focus of evaluating such meta-learning algorithms is to decouple the question of algorithm design from the question of architecture design. Nevertheless, we agree that scalability is an important criterion for evaluating meta-learning methods.
>
> Lastly, we note that a similar method to TCML (https://openreview.net/forum?id=B1DmUzWAW) reports results with a shallow miniImageNet embedding: "5-way mini-Imagenet: 45.1% and 55.2% (1-shot, 5-shot)." In future revisions of this work, we will treat this as the comparison point for a temporal-convolution-based meta-learner that employs the standard shallower architecture.
>
> -----------------------------------------------------------------------------------------------------------------------------------------
> Previous meta-learning methods applied to miniImageNet that employ the architecture of Vinyals et al. (2016):
>
> - Vinyals et al. (2016). "Matching Networks for One Shot Learning." (https://arxiv.org/abs/1606.04080)
> - Ravi & Larochelle (2017). "Optimization as a Model for Few-Shot Learning." (https://openreview.net/forum?id=rJY0-Kcll)
> - Finn et al. (2017). "Model-Agnostic Meta-Learning for Fast Adaptation of Deep Networks." (https://arxiv.org/abs/1703.03400)
> - Snell et al. (2017). "Prototypical Networks for Few-shot Learning." (https://arxiv.org/abs/1703.05175)
> - Triantafillou et al. (2017). "Few-Shot Learning Through an Information Retrieval Lens." (https://arxiv.org/abs/1707.02610)
> - Li et al. (2017). "Meta-SGD: Learning to Learn Quickly for Few-Shot Learning." (https://arxiv.org/abs/1707.09835)
>
> -----------------------------------------------------------------------------------------------------------------------------------------
> Previous meta-learning methods applied to miniImageNet that make use of an alternative architecture:
>
> - Munkhdalai & Yu (2017). "Meta Networks." (https://arxiv.org/abs/1703.00837)
>     - Use a "CNN [with] 5 convolutional layers, each of which is a 3×3 convolution with 64 filters, followed by a ReLU non-linearity, a 2×2 max-pooling layer, a fully connected (FC) layer, and a softmax layer" (App. A).
>
> - Mishra et al. (2017). "Meta-Learning with Temporal Convolutions." (https://arxiv.org/abs/1707.03141)
>     - Use "14 layers of 4 residual blocks [each with] a series of [three] convolution layers followed by a residual connection and then a 2×2 max-pooling operation" (App. C).
>
> - Sung et al. (2017). "Learning to Compare: Relation Network for Few-Shot Learning." (https://arxiv.org/abs/1711.06025)
>     - On top of the standard Vinyals et al. (2016) architecture, add a relation module that "consists of two convolutional blocks and two fully-connected layers. Each of convolutional block [sic.] is a 3×3 convolution with 64 filters followed by batch normalisation, ReLU non-linearity and 2×2 maxpooling... The two fully-connected layers are 8 and 1 dimensional, respectively." (Section 3.4).

---

### Author Response · Authors · 2018-01-06
**Main response to reviewers**

We thank the reviewers for their constructive feedback! We have updated the paper as follows:

- Section 3.1: We have clarified that every iterate of truncated gradient descent is a mode of an implicit posterior, and thus the gradient descent procedure during fast adaptation does not need to be taken to convergence.

- Figure 5 & Section 5.1: We have substantially revised the toy example in Section 5. We show that the interpretation of the method as hierarchical Bayes makes it practical to directly sample model parameters in a sinusoid regression task, and that there is increased uncertainty over model parameters when the datapoints for the task are less informative.

- Everywhere: We have incorporated minor reviewer comments regarding typos, clarifications, etc.

We respond to individual comments in direct replies to the reviewers’ comments.

---

### Decision · Program_Chairs · 2018-01-29
**ICLR 2018 Conference Acceptance Decision**

**Decision:**

Accept (Poster)

**Comment:**

Pros:
+ The paper introduces a non-trivial interpretation of MAML as hierarchical Bayesian learning and uses this perspective to develop a new variation of MAML that accounts for curvature information.

Cons:
- Relatively small gains over MAML on mini-Imagenet.
- No direct comparison against the state-of-the-art on mini-Imagenet.

The reviewers agree that the interpretation of MAML as a form of hierarchical Bayesian learning is novel, non-trivial, and opens up an interesting direction for future research.  The only concerns are that the empirical results on mini-Imagenet do not show a particularly large improvement over MAML, and there is no direct comparison to the state-of-the-art results on the task.  However, the value of the new perspective on meta-learning outweighs these concerns.